# FEATURES EMERGE AS DISCRETE STATES: THE FIRST APPLICATION OF SAES TO 3D REPRESENTATIONS

**Albert Miao**[1]  **Chenliang Zhou**[1,*]  **Jiawei Zhou**[2]  **Cengiz Oztireli**[1]

## ABSTRACT

Sparse Autoencoders (SAEs) have found human-interpretable features in LLM activations, clarifying how LLMs transform input to output. However, they have rarely been applied outside of text, limiting explorations of feature dynamics. We present the **first application of SAEs to the 3D domain**, analyzing the features found in 53k 3D objects encoded by a state-of-the-art 3D reconstruction VAE. We observe that the model encodes discrete rather than continuous features, leading to our key finding: **the model's feature activations approximate a discrete state space, driven by phase-like transitions**. Through this state space framework, we address three otherwise unintuitive behaviors — the preference for positional encoding features, the sigmoidal relationship between feature ablation and reconstruction loss, and the bimodal distribution of phase transition points. This final observation suggests the model **redistributes superposition interference to prioritize the high-importance features**. Our work not only catalogs and explains unexpected feature dynamics, but also provides a framework to explain the model's learning dynamics. The code is available at https://feature3d.github.io/Dora-SAE/.

## 1 INTRODUCTION

Interpretability research has recently focused on feature decomposition: the process of translating internal model activations into human-readable concepts. To this end, studies have used sparse autoencoders (SAEs) as a dictionary-learning tool (Bricken et al. 2023). These publications find semantically interpretable pipelines in foundational models for a myriad of tasks, including arithmetic (Lindsey et al. 2025), protein characteristics (Garcia & Ansuini 2025), and image-text relationships (Yan et al. 2025). We cite further studies in Appendix A.

The success of these methods suggests that internal activations function as a concept space, where independent feature vectors point in different directions (Elhage et al. 2023). Furthermore, models can include a number of feature vectors far greater than the cardinality of the latent space through a process called superposition, at the cost of interference (Hänni et al. 2024). However, research in feature decomposition is lacking in two key areas. First, the scope of data domains has been limited — recent feature decomposition techniques, particularly SAEs, have mostly focused on text, raising questions on their generality. Second, existing research tends to empirically discuss *what* features contribute to a model's performance, rather than explore *why* or *how* these features were chosen by the model. In short, current SAE-based studies are text-focused and descriptive, rather than explanatory. We believe the field should aim for a domain-agnostic framework to explain how features are learned and organized.

To the best of our knowledge, we are the first work to apply an SAE to 3D data. We analyze 53k 3D objects sampled from Objaverse (Deitke et al. 2022) encoded with Dora-VAE (Chen et al. 2024). 3D data is well-suited for our purposes, because data points in 3D space are drawn from an unordered, continuous domain. This is a qualitatively distinct challenge to text. In addition, a) 3D data is visually obvious when detected features have semantic meaning, b) existing datasets have a variety of recognizable objects with unique semantic combinations, c) many industries that use 3D data rely on AI tools, and d) we avoid datasets that are noisy, synthetic, homogeneous, sparse, and/or constructed for toy experiments.

---

[1] University of Cambridge   [2] Stony Brook University
[*] Corresponding Author

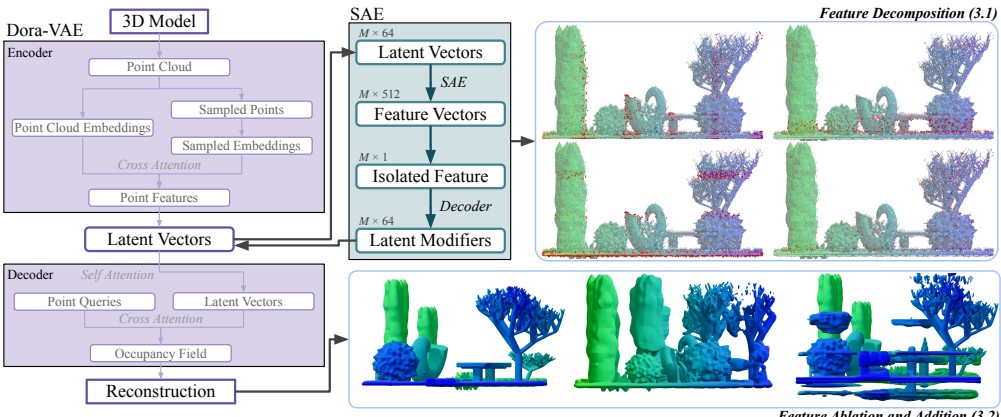

Figure 1: Our feature decomposition pipeline. Dora-VAE is a 3D reconstruction model, encoding 3D objects to $M$ latent vectors each. We apply our SAE to these latent vectors, decomposing each vector to a linear combination of features. We can visualize the effects of an individual feature by plotting the its presence in each latent vector (Section 3.1) or modifying the latent vector and observing the effects on the reconstruction (Section 3.2).

We extract human-interpretable features from latent vectors and report on their semantic meaning. We also observe that Dora-VAE learns unorthodox feature representations. Namely, positional features are represented *discretely rather than continuously*, transition points of high-impact features follow a *unimodal distribution*, and transition points of low-impact features follow a *bimodal distribution*.

Where previous work prioritized cataloging features, we study the underlying **learning dynamics**. We deconstruct the optimization step and identify two terms that independently attend to the **presence** and **identity** of individual features. The dichotomy between these terms offers explanations to the learning behaviors of Dora-VAE highlighted earlier — particularly, we suggest the unimodality of high-impact transition states is explained through the presence term, and the bimodality of low-impact transition states is explained through the redistribution of superposition interference. This framework is potentially universally applicable, intended to provide context for future interpretability work to discuss how concepts form a discrete state space.

We substantiate our framework through verifying experiments. First, we substantiate our learned features through targeted feature intervention. We show the clear semantic effect a feature has on a reconstructed output. Second, we observe and discuss several counter-intuitive behaviors that are explained by our framework. We perform 848k independent feature interventions across a set of 53k 3D models to observe patterns of changes in loss.

Ultimately, we provide the first application of SAEs to 3D data, and explain unusual properties of the feature space through a novel framework. In future work, we hope to evaluate the universality and consistency of our framework on different models and modalities, as well as further investigate the feature dynamics of models in the 3D domain.

## 2 FEATURE DECOMPOSITION PRELIMINARIES

We describe a model as the composition of functions $f : \mathbf{x} \mapsto \mathbf{z}$ and $g : \mathbf{z} \mapsto \mathbf{y}$, where $\mathbf{z}$ is a latent vector. Given a dataset $\{(\mathbf{x}_i, \mathbf{y}_i)\}_{i=1}^{N}$, the training objective is to optimize parameters $\theta_f$ and $\theta_g$ by minimizing a loss function $\mathcal{L}$ over the dataset:

$$\min_{\theta_f, \theta_g} \sum_{i=1}^{N} \mathcal{L}\left(g(\theta_g; f(\theta_f; \mathbf{x}_i)), \mathbf{y}_i\right) \tag{1}$$

Several works have suggested theoretically (Bengio et al. 2013) and empirically (Elhage et al. 2022) that latent representations in both humans and models can be viewed sets of semantic ideas

— borrowing terminology proposed by Kim et al. (2018), we denote these ideas as a set of vectors $\mathbf{E}$, and the space spanned by these vectors as $E$. These works suggest that $\mathbf{z} \in E$ and that $\mathbf{z}$ can be decomposed to a linear combination of vectors from $\mathbf{E}$:

$$\mathbf{z} = f(\theta_f; \mathbf{x}) = \mathbf{E}(\theta_f)^T \boldsymbol{\alpha}(\theta_f; \mathbf{x}) \quad \text{where} \quad \mathbf{E} = [\mathbf{e}_1, ..., \mathbf{e}_n] \subset E, \quad \boldsymbol{\alpha} = (\alpha_1, ..., \alpha_n)^T \quad (2)$$

Abstractly, we state that a model's latent space interprets a given input $\mathbf{x}$ as a set of scalars $\boldsymbol{\alpha}$ modifying a set of learned features $\mathbf{E}$. We refer to $\alpha_j$ as the *presence* of feature $j$ and $\mathbf{e}_j$ as the *identity* of feature $j$.

Even if an input $\mathbf{x}$ is out of the domain of the training dataset, the model still attempts, and likely fails, to frame the input in these features. For example, image adversarial attacks use noise that are completely out-of-distribution, but a classification model must still estimate the presence of each feature $\mathbf{e_j}$ (Gorton & Lewis 2025). If the resulting feature presences are similar to an in-distribution input, the image is misclassified.

Recent LLM studies (Bricken et al. 2023) use a sparse autoencoder (SAE) to approximate this decomposition with the assumption that $\boldsymbol{\alpha}$ is sparse; that is, the number of feature vectors in $\mathbf{E}$ (i.e. the dictionary size) is large compared to the number of features needed to represent a vector in $E$. Given a collection of input $\mathbf{x}$ and their corresponding latent vectors $\mathbf{z}$, we attempt to approximate $\boldsymbol{\alpha}$ and $\mathbf{E}$ through the following parametrization, known as a BatchTopK SAE (Bussmann et al. 2024):

$$\text{For fixed } \theta_f : \quad \boldsymbol{\alpha}(\theta_f, \mathbf{x}) \approx \text{Enc}(\mathbf{z}) \qquad \text{where} \quad \text{Enc}(\mathbf{z}) = \text{TopK}(\mathbf{W}^{Enc}\mathbf{z} + \mathbf{b}^{Enc})$$
$$\mathbf{E}(\theta_f) \approx \mathbf{W}^{Dec} \qquad \text{where} \quad \hat{\mathbf{z}} = \mathbf{W}^{Dec}\text{Enc}(\mathbf{z}) + \mathbf{b}^{Dec} \tag{3}$$

where TopK selects the top $k$ nonzero values across the batch. The linear weight matrix $\mathbf{W}^{Dec}$ approximates the set of features $\mathbf{E}$, forming an overcomplete dictionary. Thus, $\text{Enc}(\mathbf{z})$ is a sparse representation of $\mathbf{z}$ using $\mathbf{W}^{Dec}$ as the set of feature vectors. We train with standard reconstruction loss, alongside an auxiliary loss based on the reconstruction from dead features — $\hat{\mathbf{z}}_{dead}$ is the reconstruction using only dead features, and $\beta$ is a scalar hyperparameter (Gao et al. 2025).

$$\mathcal{L}(\theta_{SAE}) = \mathcal{L}_{recon}(\mathbf{z}, \hat{\mathbf{z}}) + \beta \mathcal{L}_{recon}(\mathbf{z}, \hat{\mathbf{z}}_{dead})$$
$$\mathcal{L}_{recon}(\mathbf{z}, \hat{\mathbf{z}}) = \sum_i ||z_i - \hat{z}_i||_2^2 \tag{4}$$

## 2.1 THE LEARNING DYNAMICS OF FEATURE DECOMPOSITION

Existing applications of SAEs to LLMs (Templeton et al. 2024) have shown extracted feature vectors that have clear semantic meaning. However, this begs the question: **Why were these features learned by the model?**

We examine the learning dynamics of $\boldsymbol{\alpha}$ by deconstructing the gradient step. Optimization of $\theta_f, \theta_g$ is performed using gradient descent:

$$\theta_f \leftarrow \theta_f - \eta \nabla_{\theta_f} \mathcal{L}, \quad \theta_g \leftarrow \theta_g - \eta \nabla_{\theta_g} \mathcal{L}, \tag{5}$$

We take the gradient of $\mathcal{L}_{total}$ with respect to $\theta_f$, substituting $\mathbf{z}$ with our definition from Equation 2.

$$\frac{\partial \mathcal{L}}{\partial \theta_f} = \frac{\partial \mathcal{L}}{\partial \hat{\mathbf{y}}} \cdot \frac{\partial \hat{\mathbf{y}}}{\partial \mathbf{z}} \cdot \frac{\partial \mathbf{z}}{\partial \theta_f} \quad \text{where} \quad \frac{\partial \mathbf{z}}{\partial \theta_f} = \sum_{j=1}^n \left( \frac{\partial \alpha_j(\theta_f; \mathbf{x})}{\partial \theta_f} \cdot \mathbf{e}_j(\theta_f) + \alpha_j(\theta_f; \mathbf{x}) \cdot \frac{\partial \mathbf{e}_j(\theta_f)}{\partial \theta_f} \right) \tag{6}$$

We see that parameters are updated with respect to features in two ways: for each feature $j$, $\nabla_{\theta_f} \alpha_j$ modifies the magnitude and frequency it fires, and $\nabla_{\theta_f} e_j$ modifies the information carried. This framework allows us to explain the following behaviors, through the lens of feature-based learning dynamics.

**The model learns features with discrete, state-like presences rather than those with a continuous spectrum of presences**. We see that the $\alpha_j$ term controls the learning rate for $\nabla_{\theta_f} e_j$, suggesting that models prefer to learn features $e_j$ that naturally have high $\alpha_j$. (*Section 4*)

**High-impact features have phase transition points that form a centered, unimodal distribution**. A transition point is where $\frac{\partial \mathcal{L}}{\partial \mathbf{z}}$ is highest, greatly affecting $\nabla_{\theta_f} \alpha_j$. The model is incentivized to represent the presence of each feature in states that are far from this point. (*Section 5.1*)

**Low-impact features have phase transition points that form a symmetric bimodal distribution**. When condensing high-dimensional information into low-dimensional space, feature presences are affected by interference from superposition. While transition points would typically be centered as above, the model minimizes damaging superposition by perturbing presences of low-impact features. (*Section 5.2*)

## 3 APPLICATIONS TO 3D RECONSTRUCTION

To verify our analysis, we apply an SAE to Dora-VAE (Chen et al. 2024). Dora-VAE is a Variational Autoencoder (VAE) that encodes point clouds $\mathbf{P_d}$ sampled from 3D models to condensed latent representations. These representations are then queried for diffusion-based reconstruction of the initial geometry. Rather than a global latent for each shape, Dora-VAE selects a set $\mathbf{P_C}$ of $M$ poin features from $\mathbf{P_d}$ using furthest point sampling (FPS), which is passed through several cross-attention layers alongside $\mathbf{P_d}$. This forms a set $\mathbf{C}$ of processed point features.

$$\mathbf{P_C} = \text{FPS}(\mathbf{P_d})$$
$$\mathbf{C} = \text{CrossAttn}(\text{PosEnc}(\mathbf{P_C}), \text{PosEnc}(\mathbf{P_d})) \tag{7}$$

We take the provided Dora-VAE network, pretrained on a subset of Objaverse, and encode 53k objects from Objaverse-XL. These encodings form a dataset of pre-KL embedding network states. The number of latents in $\mathbf{C}$ is determined by the number of points initially sampled; we record a set for $M = 4096$, where each pre-embedding is size 128.

After encoding, for each pre-embedding, Dora-VAE isolates a mean $\boldsymbol{\mu}_i \in \mathbb{R}^{64}$ and variance $\boldsymbol{\sigma}_i \in \mathbb{R}^{64}$ by chunking.

$$\mathbf{C} = \{(\boldsymbol{\mu}_i, \boldsymbol{\sigma}_i)\}_{i=1}^{M}$$

Thus, $\forall i \in \{1, 2, ..., M\}, j \in \{1, 2, ..., 64\}$, the KL embedding is:

$$z_{i,j} = \mu_{i,j} + \sigma_{i,j} \cdot \epsilon \tag{8}$$

where $\epsilon \sim \mathcal{N}(0, 1)$. This embedding is fed through the decoder before querying for occupancy. Here, the term *latent* with respect to Dora-VAE will refer to the network state post-KL embedding.

### 3.1 SAE ON DORA-VAE

The dataset for our SAE is constructed from these recorded pre-embeddings. Each epoch, through KL, we sample each recorded pre-embedding for new latents. Our latent space is thus extremely well-defined, as each epoch of training has 217 million newly sampled latents. In addition, since these latents are point cloud features initially downsampled from $\mathbf{P_d}$, each latent will correspond to a point of the initial point cloud sample. This relationship allows us to interpret a feature based off of the position or structure of points with high presence for that feature.

We train our BatchTopK SAE with $M = 4096$, codebook size $n = 512$, threshold $k = 8$, and $\beta = 0.125$. We use a batch size of 327680 latents, randomly selected regardless of which 3D model produced each latent. We use the Adam optimizer with an initial learning rate of 1e-3 and train for ten epochs. The model was trained on a single A100 and took 2 hours to train. We also present metrics for variations on codebook size and threshold in Appendix B.1.

We highlight the qualitative performance of our feature extraction in Figure 2. For each encoded 3D object, we obtain a set of latents $\{\mathbf{z}_i\}_{i=1}^{M}$, and $\boldsymbol{\alpha}_i$ for each latent by passing it through the SAE. We plot the $M$ latents as points from their initially sampled positions $\mathbf{P_d}$. Finally, to examine a feature $j$, we color each point $i$ of each latent based off the presence $\alpha_{i,j}$.

Most features display positional information along a single axis. Notably, features appear state-like, and store information in a binary manner. Features emerge at striped intervals across models in a manner akin to positional encoding, suggesting latents form a discrete representation. In other words, a feature doesn't have a range of possible values (*"As feature $j$ increases, the point travels*

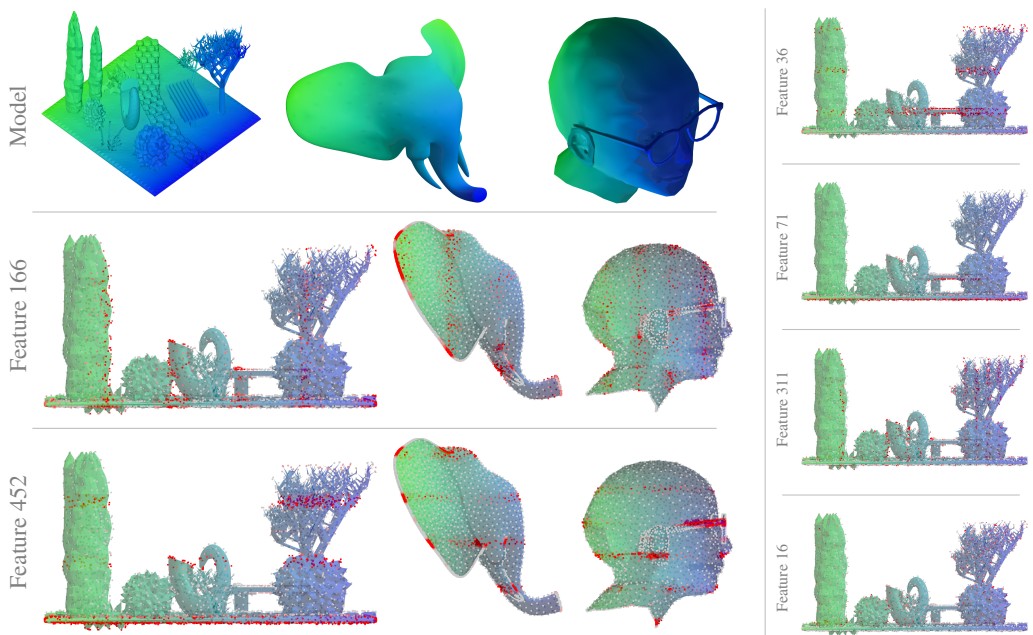

Figure 2: Overview of Dora-VAE features. Each point's color represents the presence of feature $j$ at that point. Visually, most features congregate in stripes along a single axis. This suggests that continuous position is represented by a set of discrete elements, which each activate in separate regions. We show further examples in Appendix C.

*further along the axis"*) — instead, a feature tends towards one of two states (*"If feature $j$ is present, it is within this region"*). We discuss the discretization of features further in Section 4.

These positional features are highly visually interpretable due to the 3D medium. We see that such features are applicable across all models, and activate with significant sensitivity and specificity. Some features, although they are highly present across the model, have meanings that are difficult to interpret through observation. We can instead intervene on these features to determine their purpose and importance.

## 3.2 FEATURE ABLATION AND ADDITION

It is possible these features are simply vestiges of correlations between the sampled points; points that share close coordinates may simply propagate similarly across the encoder. To disprove this, and demonstrate these features are meaningful internal representations, we examine the downstream effects of modifying latents along feature axes.

We intervene on features through ablation and addition based on SAE decoder weights. In our pipeline, inputs are encoded by Dora-VAE to a set of latent vectors $\{\mathbf{z}_i\}_{i=1}^M$. We recall Eq. 2; to visualize the effect of modifying feature $j$ on the reconstruction, we want to approximate a modified set of latents such that:

$$\mathbf{z}_i' = \mathbf{E}^T \boldsymbol{\alpha}_i' \quad \text{where} \quad \boldsymbol{\alpha}_i' = \boldsymbol{\alpha}_i, \alpha_{i,j}' = (1-t) \cdot \alpha_{i,j} \tag{9}$$

where $(1-t)$ is the proportion of the original presence, set externally. Rather than rely on the reconstruction provided by our SAE, we modify the latents with the decoder weight for feature $j$.

$$\forall \mathbf{z}_i \in \{\mathbf{z}_i\}_{i=1}^M \begin{cases} \text{Ablation:} & \mathbf{z}_i' \approx \mathbf{z}_i - t \cdot \text{Enc}(\mathbf{z}_i)_j \mathbf{w}_j^{dec} \\ \text{Addition:} & \mathbf{z}_i' \approx \mathbf{z}_i + \alpha_j' \mathbf{w}_j^{dec} \end{cases} \tag{10}$$

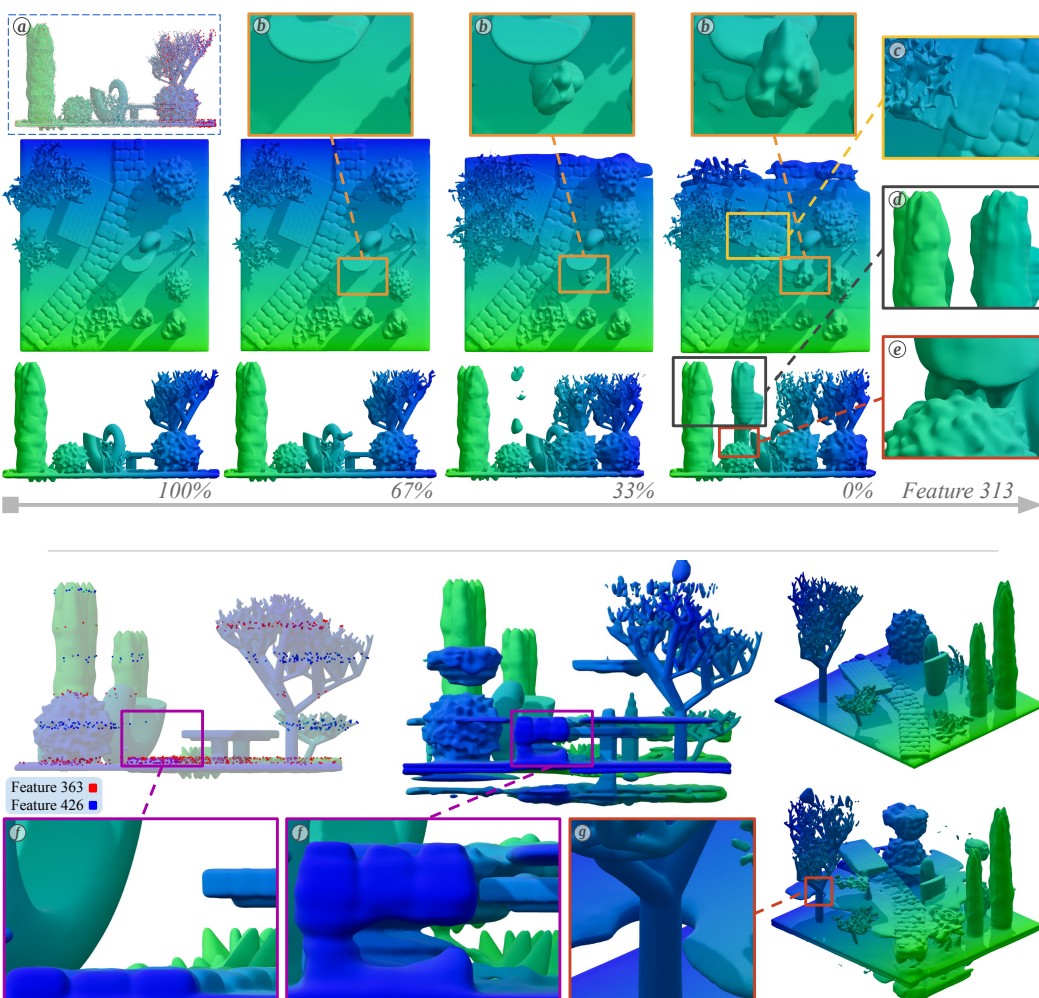

Figure 3: *Top*: Example ablation on feature 313. ⓐ This feature primarily attends to points on the positive end of the $z$-axis. As the feature is removed, shapes disappear and appear spontaneously, rather than moving along the object ⓑ. This suggests the feature represents a discrete region in space, rather than a continuous range of positions. *Bottom*: Example of feature 363 ablation and feature 426 addition. Both features attend to regions along the $y$-axis. ⓕ Shapes that had their region fixed by feature 363 are moved to regions defined by feature 426. In addition, they preserve their local structure. See Appendix B.2.

During ablation, the portion of feature $j$ removed is scaled by value $t$; here, $t = 1$ implies the feature is completely removed. During addition, feature $j$ is added by amount of a manually set $\alpha'_j$. Each modified set of latents is passed through the the Dora-VAE decoder to be compared to the original model. We record the mean squared error (MSE) of the decoded reconstruction.

The top of Figure 3 demonstrates an ablation of feature 313. As shown by the red points in ⓐ, this feature primarily attends to points on the positive end of the z-axis, with a small region on the negative end. When the feature is ablated, shapes whose position relied on it are rendered elsewhere — this is indicative of a causal relationship between this feature and the shape's position. In addition, as shown in ⓑ, rendered points appear spontaneously, rather than moving across the model. This again suggests that features represent discrete states, and presences do not have a continuous range of information.

At the bottom of Figure 3, we demonstrate an ablation of feature 363 alongside an addition of feature 426. Rather than applying a constant presence of feature 426 on all latents, we instead replace every

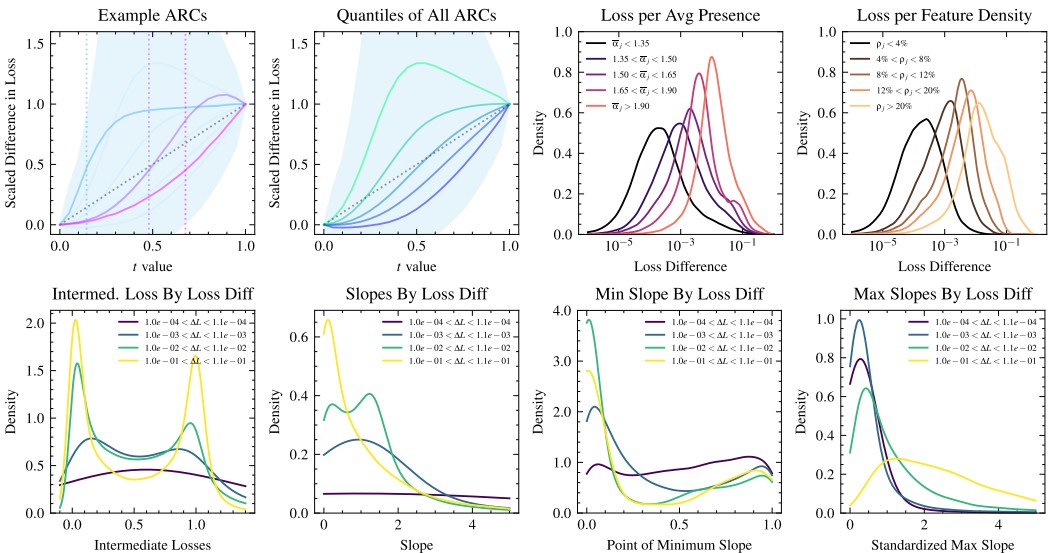

Figure 4: *Top left*: Examples and quantiles of ARCs. ARCs display an almost sigmoidal behavior, with a varied transition point. *Top right*: Correlations of loss with average value and feature density. *Bottom*: Various experiments to demonstrate discretization. ARCs will typically have stagnant MSE near the beginning and end, and change most rapidly in a small interval.

presence of feature 363 with an equal presence of feature 426.

$$\forall \mathbf{z}_i \in \{\mathbf{z}_i\}_{i=1}^M, \quad \mathbf{z}_i' \approx \mathbf{z}_i - \text{Enc}(\mathbf{z}_i)_{363}\mathbf{w}_{363}^{dec} + \text{Enc}(\mathbf{z}_i)_{363}\mathbf{w}_{426}^{dec}$$

Note that, because both features 363 and 426 attend to positions on the y-axis, points displaced by the removal of feature 363 are anchored by the addition of feature 426. Rendered shapes preserve their form even after moving. We discuss further observations in Appendix B.2.

## 4 DO FEATURES SHOW STATE-BASED BEHAVIOR?

A significant portion of features in Dora-VAE are dedicated to representing the position of the latent. Intuitively, one would assume that these features should be continuous, as points are relatively uniformly distributed across 3D space. Each latent's position could be represented by only three features, with others for additional fidelity. Despite this, the model chooses to represent features discretely; if the feature has a high presence, the position is within a defined region. This method of representation is akin to binary positional encoding. Some features make the similarity more explicit by representing a set of multiple regions across an axis, rather than a single one.

To verify whether these features are truly discrete, we perform a series of systematic feature ablations over our dataset and measure the change in loss. As above, we pass the $M$ latents of 53k 3D objects through our BatchTopK SAE with $k = 8$ and codebook size 512. Each 3D object thus has a set of presence vectors $\{\boldsymbol{\alpha}_i\}_{i=1}^M$ where each $\boldsymbol{\alpha_i} \in \mathbb{R}^{512}$ shows the presences of 512 features and has 8 nonzero values on average. For each 3D object, we randomly select 16 features to intervene on, preferring features that are present in more latents. We perform each ablation with $t \in \{0.00, 0.05, 0.10, ..., 1.0\}$, recording the MSE of the decoded reconstruction for each $t$. We evaluated 848k ablations in total.

The model's response to feature ablation displays interesting recurring behaviors. Given our set $\{\boldsymbol{\alpha}_i\}_{i=1}^M$, we define the feature density of feature $j$ as $\frac{1}{M}\sum_{i=1}^M \mathbf{1}\{\alpha_{i,j} \neq 0\}$, and the average presence of feature $j$ as $\frac{1}{M}\sum_{i=1}^M \alpha_{i,j}$. We also define the impact $\Delta L$ of an ablation as the difference in MSE between $t = 0$ and $t = 1$. Figure 4 shows kernel density estimations (KDEs) for these properties of each feature ablation. We note that our set of ablations shows a wide variety of im-

pact, feature density, and average presence. In addition, impact is positively correlated with both the feature density and average presence.

We also plot several ablation-response curves (ARCs). Each curve represents a single ablation, and shows the change in MSE as $t$ increases. We normalize MSE such that the plotted error at $t = 0$ is 0, and at $t = 1$ is 1. We also record the transition point of an ablation as the value of $t$ when the normalized MSE is 0.5. Note that the ARCs do not show a linear relationship between change in latents and MSE. Rather, they exhibit variable curvature, with two inflection points — initial changes in loss are below our projected linear growth, then accelerate at the transition point, before again slowing down.

We find ARCs with greater impact exhibit more discrete behavior. To demonstrate this, we group ARCs together based on $\Delta L$ and perform four experiments. First, for each group of similar $\Delta L$, we plot a KDE of all normalized MSE for $0.05 \leq t \leq 0.95$. Notably, as impact increases, intermediate MSE values tend to cluster towards the initial and final MSE. Second, we analyze the maximum slope of each ARC to determine if it is an outlier in the distribution of slopes. For each ARC, we estimate slope as the difference of normalized MSE between every two consecutive $t$ values. We then z-score the greatest slope of each ARC relative to the distribution formed by all slopes in the group, and plot the average for each group. As the $\Delta L$ of an ARC increases, the greatest slope of the ARC trends further from the group distribution, suggesting the transition point is more well defined as impact increases. Third, we find the point of flattest slope for each ablation, record the value of $t$, and plot a KDE of these values. We see that points near the beginning and end have typically the flattest slope. Finally, we note that the distribution of slopes across ARCs leans further left as $\Delta L$ increases, showing that ARCs are typically flatter, with sharper jumps, when impact increases.

The discretization of features can be explained through the learning dynamics defined in Section 2.1. The signal to the identity of feature $j$, $\nabla_{\theta_f} \mathbf{e}_j$, is scaled by the presence $\alpha_j$. Thus, the identity of feature $j$ is most influenced when $\alpha_j$ is high, while, when $\alpha_j$ is low, the signal to identity is diluted by other signals for features with higher presence. We further discuss this intuition in Appendix B.3.

## 5   THE BIMODALITY OF TRANSITION POINTS

If we interpret feature activations as a discrete state space with distinct phase transitions, we can follow up by investigating when these phase transitions occur. We plot KDEs of two properties — transition points and the points of greatest slope — for all ARCs in Figure 5. Surprisingly, both distributions are bimodal.

We further investigate this behavior by again grouping ARCs by $\Delta L$ and plotting the KDEs of transition points and points of greatest slope for each group. From this figure, we see that the transition points of high-impact ARCs form a unimodal distribution around the center ($t \approx 0.5$), while the transition points of low-impact ARCs form a bimodal distribution roughly symmetric about this center.

We then further group ablations based on which feature $j$ is removed and repeat the same investigations as above. Again, for high-impact ablations, each feature has a peak near the center. However, low-impact ablations are no longer bimodal, and instead there is instead a single peak that strays from the center. Some fall closer to the beginning, while others are nearer the end. Thus, it is only when all ARCs are aggregated together, regardless of feature, that we observe a bimodality of transition points in low-impact features.

### 5.1   UNIMODAL TRANSITION POINTS OF HIGH-IMPACT ABLATIONS

We explain the distribution of high-impact transition points using the learning dynamics defined in Equation 6. Because our feature activations, especially high-impact ones, approximate discrete behavior, we can consider a feature to be *on* (high presence) or *off* (low presence). During the gradient step, the effect on $\nabla_{\theta_f} \alpha_j$ is scaled by $\frac{\partial \mathcal{L}}{\partial \mathbf{z}}$. This gradient, by definition, reaches a peak near the transition point. Because of the rapid change in loss associated with the transition point, $\nabla_{\theta_f} \alpha_j$ is incentivized to adjust $\alpha_j$ such that both on and off states of the feature are at a distance from the transition point. The magnitude of $\frac{\partial \mathcal{L}}{\partial \mathbf{z}}$ at the transition point is not necessarily equivalent for when

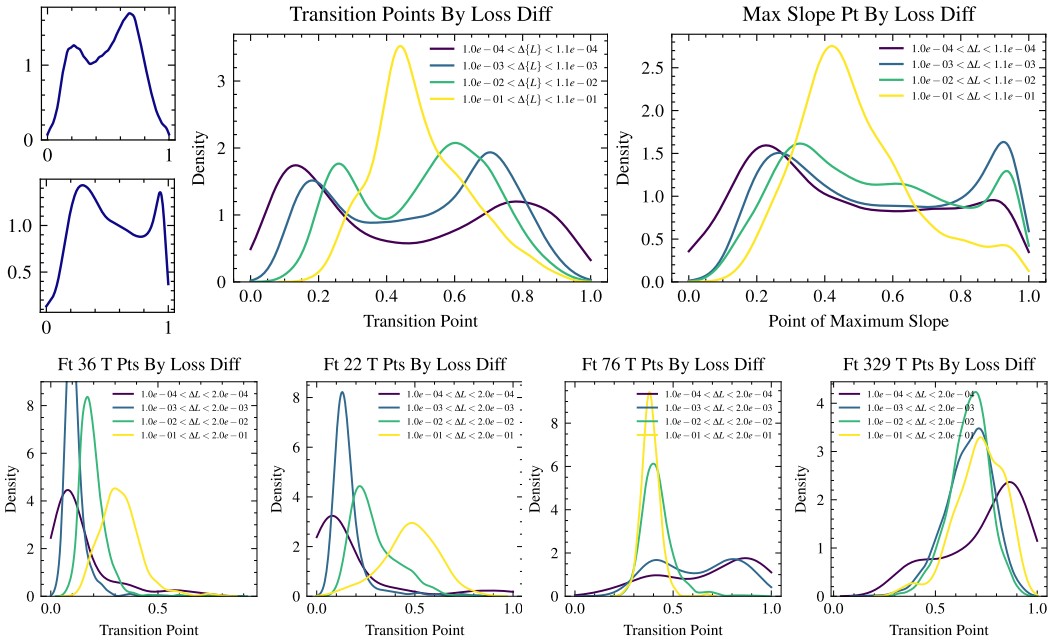

Figure 5: *Top*: Distributions of transition points and points of maximum slope over all ARCs, grouped by $\Delta L$. ARCs with greater $\Delta L$ form a unimodal distribution of transition points centered about $t \approx 0.5$, while those with less $\Delta L$ have a symmetric, bimodal distribution. *Bottom*: Distributions of transition points of individual features, grouped by $\Delta L$. ARCs with greater $\Delta L$ still form a somewhat centered, unimodal distribution, but those with less $\Delta L$ now also unimodal, further from the center. This polarization across individual features causes the bimodal distribution when considered in aggregate.

feature $j$ is on and off — however, over many different transition points, we can assume it is roughly symmetric. Thus, the overall distribution of high-impact transition points is located at the center.

## 5.2 BIMODAL TRANSITION POINTS OF LOW-IMPACT ABLATIONS

The bimodality of low-impact transition points is a more complex property. When examining individual features, the distribution of low-impact transition points forms a unimodal peak that drifts away from the center. This behavior is not caused by a weaker $\frac{\partial \mathcal{L}}{\partial \mathbf{z}}$ at the transition point, which would've only increased the variance of the distribution. Instead, the peak itself is offset to the left or right — as if a polarizing effect drives transition points away from the center as $\Delta L$ decreases.

We speculate that this effect is caused by a variable offset applied to the feature presence, and provide a visual aid in Figure 6. We suggest that as impact decreases, the larger this offset becomes. In this way, the transition point is moved earlier or later in the ablation. This effect does not make the feature presence estimation less accurate; again, in that case, we would see the low-impact distribution have greater variance, but retain the same center. Rather, the peak itself moves. We hypothesize that this offset is caused by the model learning to redistribute interference from superposition.

Superposition occurs when high-dimensional features are constrained to a low dimensional space, causing interference between features in previously distinct dimensions. (Elhage et al. 2022). We show another visual aid in Figure 7. Suppose the model has determined to represent a latent through high-impact feature 1, low-impact feature 2, and a set of several other features $*$. We refer to the presences as $\alpha_1$, $\alpha_2$, and $\alpha_*$. As shown at the top of Figure 7, due to superposition, the features $*$ add interference in the direction of $\alpha_1$ and $\alpha_2$. We suggest that the model selects features $*$ such that feature 2, a low-impact feature, is affected by this superposition more than feature 1, a high-impact feature. This causes the offset shown at the bottom of Figure 7 — while the transition point of feature 1 is still centered, the transition point of feature 2 is moved.

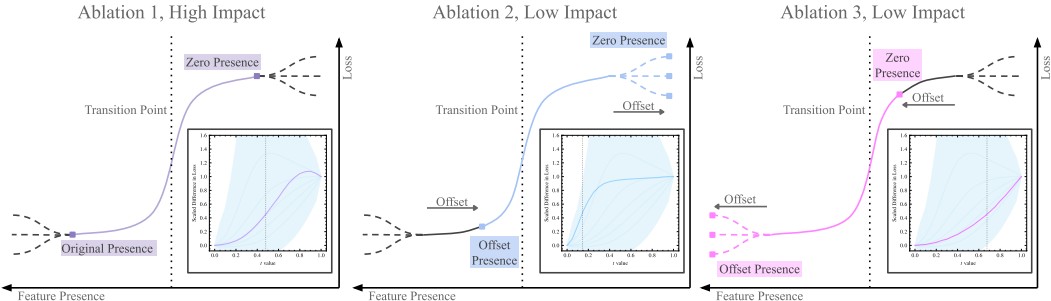

Figure 6: *Left*: A visual aid of a high-impact ARC. As feature presence is ablated, the loss (MSE) increases. It is fastest when passing through the transition point, then stagnates. *Center and Right*: A visual aid of two low-impact ARCs. We suggest that low-impact ARCs are affected by a polarizing offset. If the presence is decreased by the offset, the relative transition point is moved left. If increased, the point is moved right.

Even though $\Delta L$ has correlations with other variables, they are not strong enough to cause a third variable contamination that affects SAE estimations of presences (see Appendix B.4). Thus, we can reasonably conclude the offset is present even before the latent is passed to the SAE. In other words, we speculate the model itself is actively preserving high-impact features by passing interference to features that are lower impact.

Previous works have shown that a model learns a set of features that minimizes interference from superposition during training. However, this work suggests that a model can redistribute interference from superposition *at inference time*; that not only

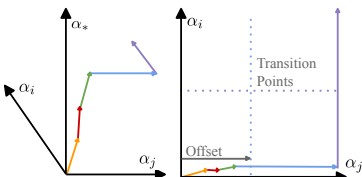

Figure 7: Superposition interference affects feature presences. We suggest this offsets low-impact features, shifting the relative transition point.

does the model select the most important features with respect to the dataset, but it dynamically estimates the importance of features *input to input*.

## 6 FUTURE WORK AND CONCLUSION

These results are motivating for feature learning dynamics research, but further work is necessary to generalize and substantiate our findings. First, by validating our observations across other domains (text, image) and models (PointNet++ (Qi et al. 2017), LION (Zeng et al. 2022)), we can confirm our results and broaden the scope of our work. Second, examining Dora-VAE with circuit detection techniques (Ameisen et al. 2025) and extracting an attribution graph may show how features flow through the architecture and form discrete patterns. Third, probing gradients of toy models, similar to Elhage et al. (2022), will evaluate the feature learning framework we present here. Finally, if we can identify what influences redistribution of interference among features, we can potentially perform feature decomposition at training time to develop a meta-learning module.

Our work is the first to apply an SAE to 3D data, highlighting specific discovered features and showing the causally related downstream effects. We then take advantage of the continuous and unstructured nature of the domain to investigate the model's feature decomposition, confirming that the latent space can be interpreted as a discrete, state-based feature space driven by phase transitions. We then provide a potentially general framework of feature learning dynamics that explains the unexpected discretization we observe. Finally, we explain the counter-intuitive property of the bimodality of transition points by proposing a mechanism by which the model redistributes superposition to only affect low-impact features.

AUTHOR CONTRIBUTIONS

Albert Miao designed the project, performed the experiments and analysis, and wrote the publication. All authors contributed through discussions and comments on the paper.

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

## A    FURTHER RELATED WORKS

The theory behind neural networks performing feature decomposition has seen several variations in previous years (Bengio et al. 2013), (Locatello et al. 2019). Even prior to studies of latent space decomposition, studies found interpretable axes in concept embeddings such as language vectors (Mikolov et al. 2013). However, in recent years, we can generally divide discussions of feature decomposition into two camps.

Empirical papers discuss the decomposition of features for interpretability of specific models or domains. Most notable of these are applications of SAEs and similar techniques on LLMs (Bricken et al. 2023), (Ameisen et al. 2025), (Gao et al. 2025), which followed the initial results of SAEs (Cunningham et al. 2024). These include analyses to extract internal representations of true and false statements (Marks et al. 2024), or discovering function vectors by analysing the cumulative impact of attention heads (Todd et al. 2024). Similarly, image classification and reconstruction studies have proposed new CNN decomposition methods, highlighting segmentations of the input image that led to appropriate classification (Ghorbani et al. 2019), (Zhang et al. 2021), (Fel et al. 2023b), (Fel et al. 2025), (Rao et al. 2024). Particularly, we point out Thasarathan et al. (2025), which suggests that text and image features can operate in the same feature space. This suggests that features can bridge modalities, and although it doesn't address it independently, the paper invites further investigation into the dynamics of cross-modal models. Still other papers propose variations or improvements on feature decomposition techniques (Rajamanoharan et al. 2024), (Bussmann et al. 2024) or draw comparisons between them (Fel et al. 2023a). Empirical papers typically focus on either the method of feature extraction or the application of specifically extracted features towards robustness, safety, or interpretability; however, they do not discuss how this feature space was learned by the model or generally functions, and, as said before, have left unstructured data domains relatively unexplored.

Theoretical papers discuss the structure and formation of the feature space. These papers are fewer and further between. We primarily draw on these for our framework of superposition, as these papers provided abstracted experiments on feature learning dynamics to build intuition (Elhage et al. 2022), (Elhage et al. 2023). Rarely, other works have investigated concept learning dynamics through accuracy evaluation of individual concepts at each stage of the model (Park et al. 2024). However, this body of work has been significantly abstracted away from current state-of-the-art models, relying on controlled or toy experiments.

In short, there has been a gap in establishing general dynamics of real-world feature spaces. Although feature decomposition itself has significantly improved with the advent of SAEs, the field lacks an equivalent explanation of how these feature spaces are formed. Our work addresses this absence though a thorough investigation of a real-world feature space, analyzing the overall trends in presence and identity of learned features in addition to their specific function. We also believe the gap in generalized feature space research is the result of heavy investment in LLM interpretability research. While such studies are clearly highly salient and fruitful, relationships between tokens are not explicit, and can be difficult to intuit. In contrast, unstructured and unordered data, while difficult to work worth, have clear spatial relationships that allow for intuitive interpretation. We hope to further discussion regarding properties of the feature space in general, as any insights are likely to inform improvements in transparency, robustness, and meta-training.

## B    ADDITIONAL DISCUSSION

### B.1    ADDITIONAL SAE VARIATIONS

| n | Relative $\ell_2$ ($\downarrow$) | Universality ($\uparrow$) | Dora-VAE Loss ($\downarrow$) |
|---|---|---|---|
| 256 | 0.518 / 0.366 / 0.194 | 0.420 / 0.421 / 0.420 | 0.538 / 0.300 / 0.228 |
| 512 | 0.507 / 0.355 / 0.187 | 0.293 / 0.297 / 0.295 | 0.427 / 0.409 / 0.216 |
| 1024 | 0.501 / 0.356 / 0.182 | 0.206 / 0.208 / 0.209 | 0.407 / 0.414 / 0.239 |

Table 1: SAE Variations, where threshold $k =$ 4 / 8 / 16.

In addition to the SAE trained in the main paper, we also train several variations on codebook size and threshold k to establish a general intuition regarding the effects of modifying these hyperparameters. We report the reconstruction loss of the latent, the loss produced by the reconstructed latent passed through the Dora-VAE decoder, and universality. Universality is a measurement of the similarity between features of separately trained SAEs, inspired by Fel et al. (2023a) and Bricken et al. (2023). To measure universality, for each set of hyperparameters, we train 10 identical SAEs on 10-fold subsets of the data. We then use a pairwise Procrustes alignment to align feature vectors between two trained SAEs. Finally, we report the average cosine similarity of vectors among paired models as universality.

We present our results in Table 1. Performance is as expected — as $k$ increases, the reconstruction is allowed higher fidelity, improving the reconstruction loss of both the SAE and Dora-VAE. Similarly, codebook size improves our metrics, as the dictionary of vectors becomes larger.

The same cannot be said for universality, however; as codebook size increases, features become less universal. We suspect this is due to the many possible tilings of positional encoding — as features are allowed to become more specific, there is a greater variation of feature collections that cover a similar space. In future work, we explore how codebook size affects the patterns of features learned.

## B.2 Further Feature Ablation and Addition Insights

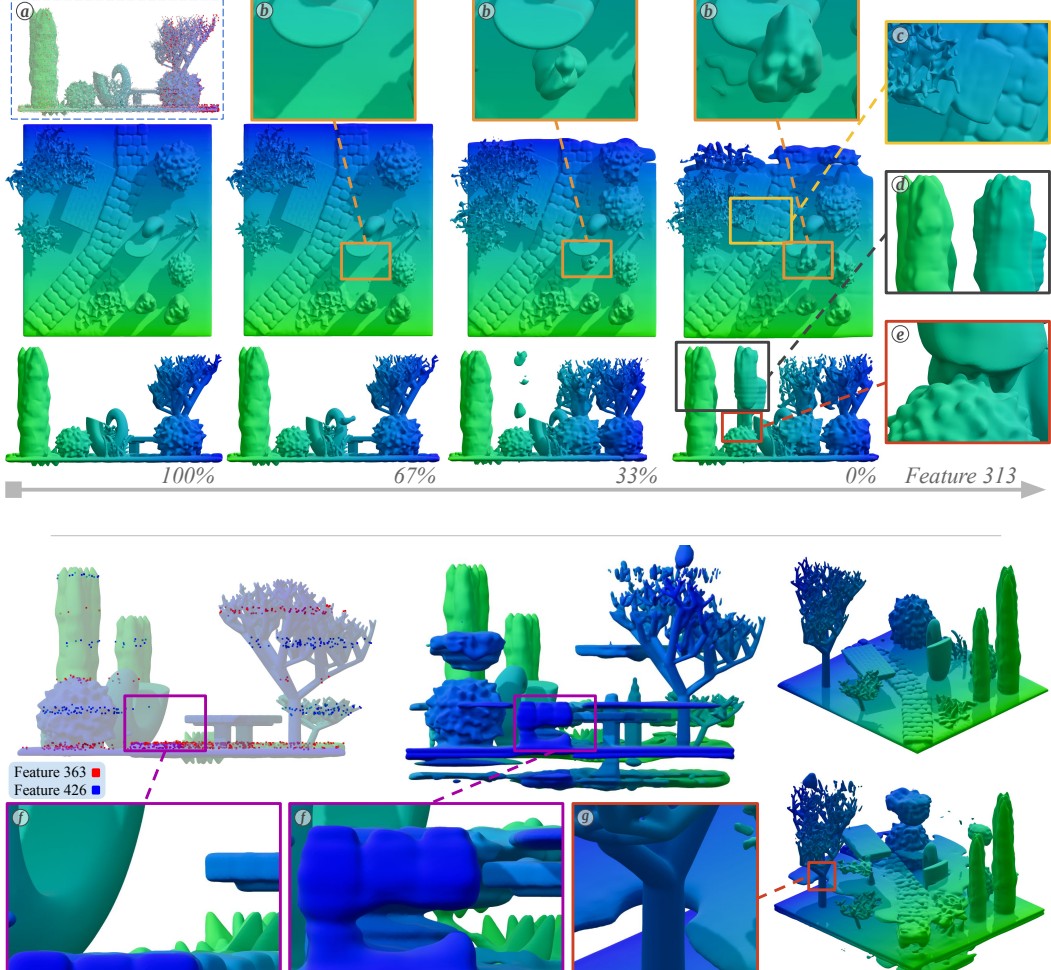

Figure 8: Reprint of Figure 3. *Top*: Example ablation on feature 313. *Bottom*: Example of feature 363 ablation and feature 426 addition.

We discuss some interesting properties in Figure 3 here. By gradually removing a single feature, we affect the reconstruction by Dora-VAE. In this case, we remove feature 313, which is responsible for points along the positive end of the z-axis, as well a small section of points towards the negative end. When we remove feature 313, these points move towards the center of the model, forming distinct shapes. We frame the discussion around several key properties.

As seen in ⓑ, the appearance of these shapes is not reflected as a continuous shift in position. Instead, points are spontaneously instantiated in their final position, with more of the shape becoming visible over time. This supports our claim that features represent discrete states.

In ⓒ, we see that points, even after shifts, can merge with their neighbors if they share compatible latents. The upper right corner of the bench, originally higher on the z-axis, merges successfully with the lower corner of the bench after transposition. This is as opposed to ⓔ and ⓖ, where the edges of the tree and bush prevent shapes that would render at that location after a shift. We suggest this is because latents in the bench have a collaborative relationship with other nearby latents, ensuring that nearby points with the same properties will attempt to merge. On the other hand, points on the edge of the tree and bush have features for non-occupancy, ensuring that other latents that would conflict with the model do not render.

ⓓ shows an example of latent redundancy within shapes. We note that, even though a section of the bush was removed, the remaining latents were able to reconstruct the local scene with strong accuracy. In addition, the new shape rendered by the shifted latents also recreates the bush with similar success. We suggest this is due to a set of redundancies within the model decoder to help preserve shapes, such that only major changes to latents can affect the model.

In the latter figure, we refer to the replacement of feature 363 with feature 426. In short, we substitute every presence of feature 363 in the model latents with a presence of equal magnitude in feature 426. As both features attend to positions on the y-axis, we see that through this substitution, we displace shapes previously located at the region marked in feature 363, and shift them to feature 426. We highlight ⓕ here — note that the cobblestone path is now elevated to a region marked by feature 426, while maintaining its shape. We use this substitution to demonstrate the independence of these features, able to affect position while still preserving the majority of the form.

### B.3 LEARNING DYNAMICS' EFFECT ON DISCRETIZATION

One might assume the discretization of features arrives solely from the interference caused by the superposition. When a model attempts to represent a large number of features in a small number of dimensions, features that appear with low presence are more likely to be construed as interference. Thus, the model learns to prefer features that only fire with a significant presence.

However, this doesn't address the effect across all features, nor does it describe the movement of parameters that brings about discretization. Notably, models prefer to superimpose features that are a) sparse and b) less important. Given that positional information is highly relevant across the entire span of inputs, it is unlikely the model would prioritize the information presented by other features at the expense of position information.

Our framework in Eq. 6 offers a possible explanation to this counter-intuitive behavior, through the second term $\alpha_j \cdot \nabla_{\theta_f} \mathbf{e}_j$. We see that the signal to the identity of feature $j$, $\nabla_{\theta_f} \mathbf{e}_j$, is scaled by the presence $\alpha_j$. Suppose we had a continuous positional feature $j$, where $\alpha_j(\mathbf{x})$ was higher and lower based on whether $\mathbf{x}$ was closer or further along a designated axis. Across $\mathbf{x}$ at a variety of positions, the identity of feature $j$ would receive the strongest signal at positions that have a high presence. Conversely, during optimization, positions with lower presence of feature $j$ have to contend with other features that have a higher presence, diluting the signal to identity. Over time, we suspect that this dynamic drives the identity of $j$ to solely consider $\mathbf{x}$ at a specific position with the highest presence, localizing the feature. This effect contributes to the discretization of all features, not just positional ones.

### B.4 CROSS-CONTAMINATION OF LOSS DIFFERENCE

One might object to the supposition that the reconstruction model itself is shifting interference from superposition, and instead suggest these shifts reflect a bias in the SAE's estimations. This objection

doesn't argue that the SAE is misestimating based on the importance of the feature (as the SAE does not optimize based off the effect of the feature on the final Dora-VAE reconstruction loss); rather, it proposes there exists a correlating property with feature importance that consistently increases the error of the SAE, causing an offset in its estimation.

Given the SAE is a simple two-layer encoder-decoder structure and that encoder and decoder weights share a high cosine similarity, if there exists a correlating property, such a feature would affect the input with a relatively linear correlation similar to the effect on the estimation observed from the output — as this feature increases/decreases, the error of the SAE increases. The two candidates with the highest potential for cross-contamination with loss difference in this manner are average presence and feature density. We investigate if these two properties drive the relationship between loss difference and transition point.

To do so, grouping ARCs by feature, we compile loss difference, average presence on active features, and feature density as dependent variables, to test their relationship with transition point location. For each feature, we perform a linear regression through OLS on these variables, as well as with their log forms, to determine which variables scale nonlinearly. We find only loss difference performs better in log. Then, again for each feature, we perform a partial $R^2$ analysis to quantify the improvement each variable has on predicting transition point. Finally, we take the average $R^2$ value for each of the investigated variables over all features in a defined subset. We repeat this investigation four times, considering subsets of features with a number of ablations $n_{abl} >= \{500, 1k, 3k, 6k\}$ ablations, and report our results in Table 2.

| $n_{abl}$ | # of Features | Log Loss Diff $R^2$ | Avg Val $R^2$ | Density $R^2$ |
|---|---|---|---|---|
| 500 | 150 | 0.188±0.104 | 0.042±0.056 | 0.025±0.027 |
| 1k | 128 | 0.181±0.105 | 0.046±0.059 | 0.023±0.025 |
| 3k | 91 | 0.183±0.116 | 0.053±0.064 | 0.020±0.018 |
| 6k | 59 | 0.188±0.119 | 0.058±0.068 | 0.017±0.017 |

Table 2: Partial $R^2$ analysis of variables contributing to transition point.

Log loss difference consistently contributes to transition point estimation more than other variables. This suggests that, rather than either average presence or feature density driving the location of the transition point, both variables' effect are likely the result of the correlation with loss difference. Thus, the offset in transition points is not caused by an artifact of the SAE.

This distinction is important. As opposed to average presence or feature density, neither the difference in loss caused by moving the latent in a chosen direction nor the location of the transition point is apparent from examining the latent individually. This suggests the model has a complex, non-linear relationship with both feature importance and relative transition point, and the correlation between these two properties is highly relevant to model behavior.

## C  FURTHER FEATURE EXAMINATIONS

We demonstrate further examinations of individual features here. Each feature, randomly selected, is highlighted across three objects, and ablated for a single object. We also give our qualitative impression of each feature's purpose. Note that some ablations have little or no effect, due to the redundancy discussed in Appendix B.2, section ⓓ.

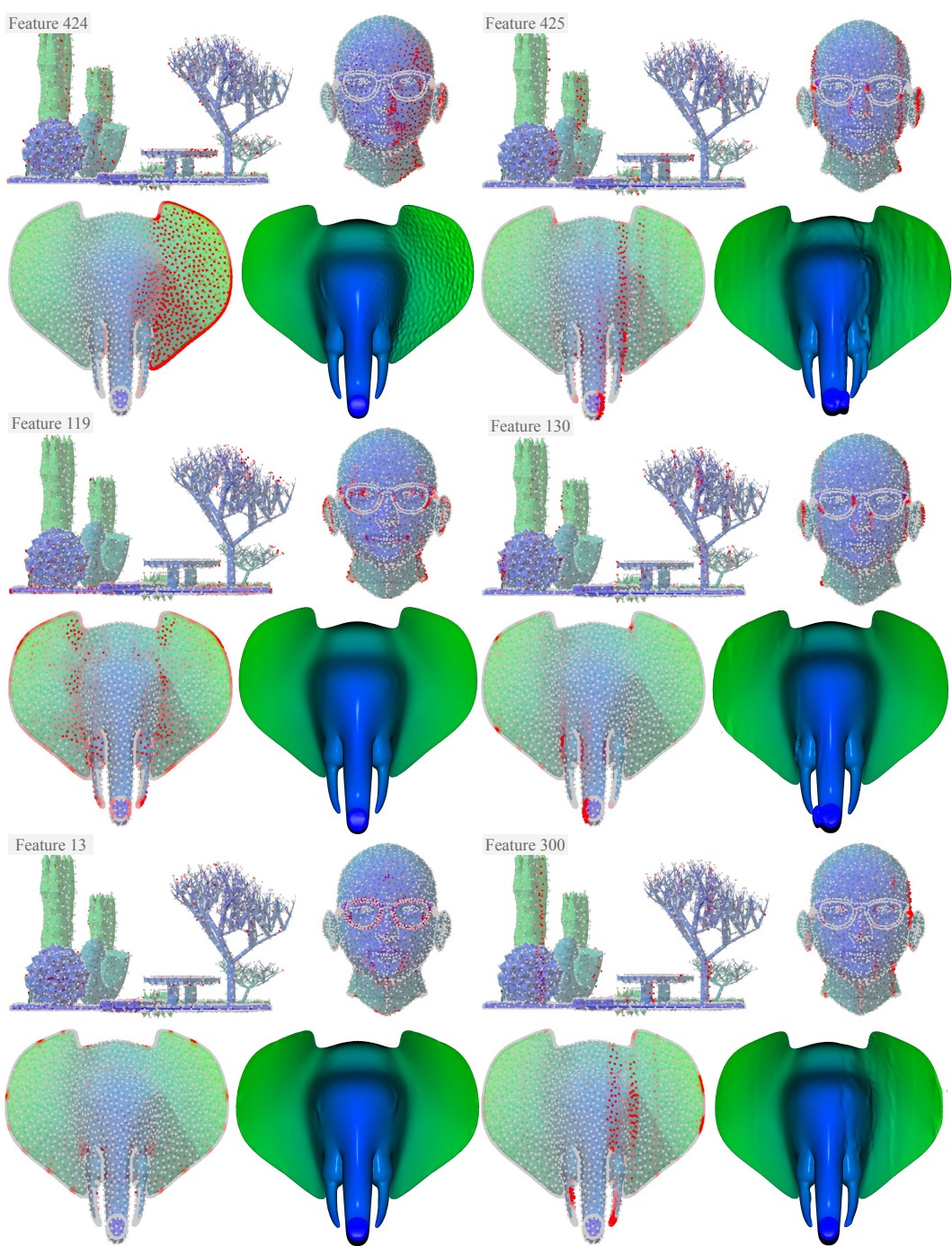

Figure 9: *Feature 424*: Smoothness of right-facing surface. *Feature 425*: x-axis encoding, right side. *Feature 119*: Unknown. *Feature 130*: Left-facing regional smoothness. *Feature 13*: Edge encoding. *Feature 300*: x-axis encoding, right side.

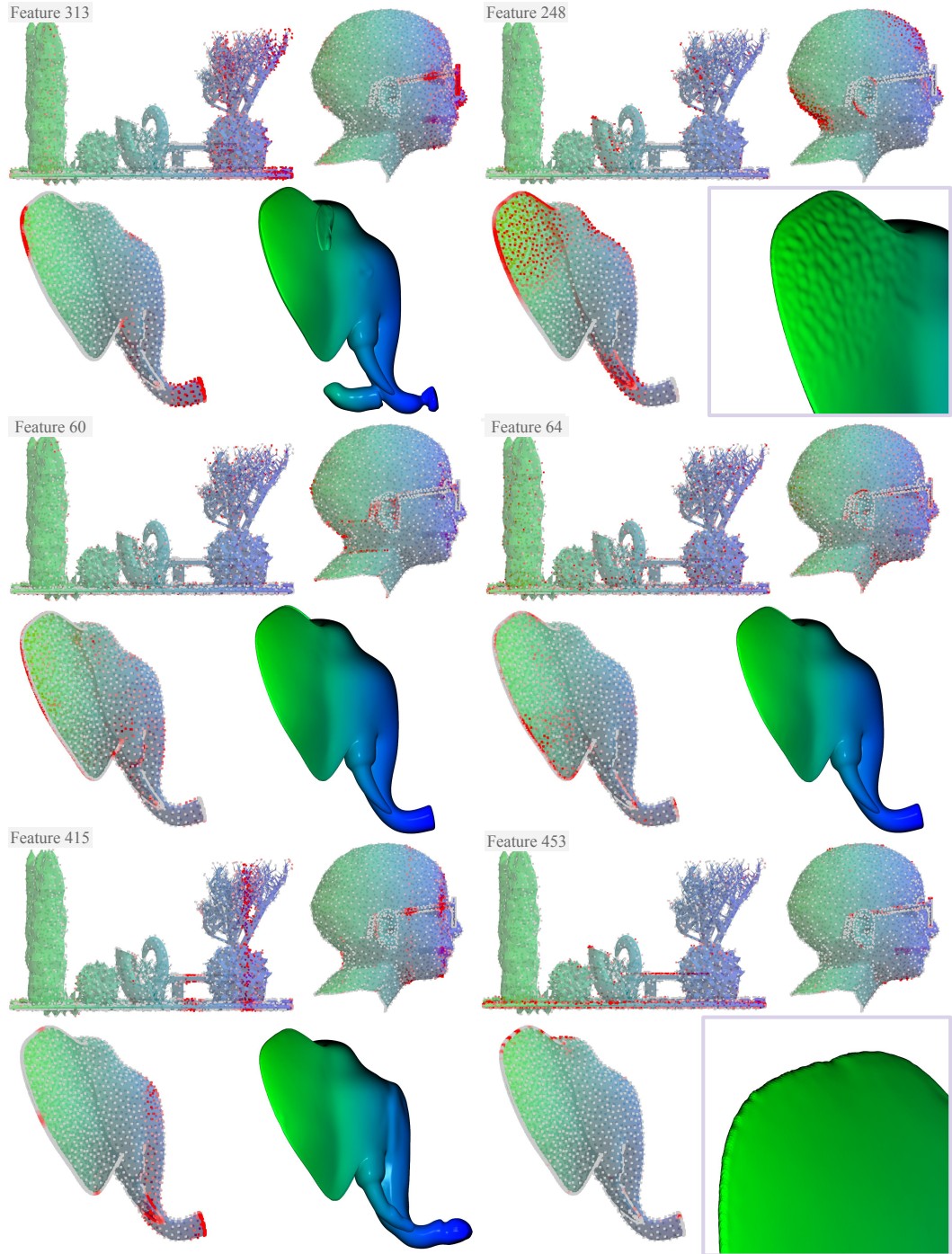

Figure 10: *Feature 313*: z-axis encoding. *Feature 248*: Smoothness of upright-facing surface. *Feature 60*: Unknown. *Feature 64*: Unknown. *Feature 415*: z-axis encoding. *Feature 453*: Upper edge encoding.

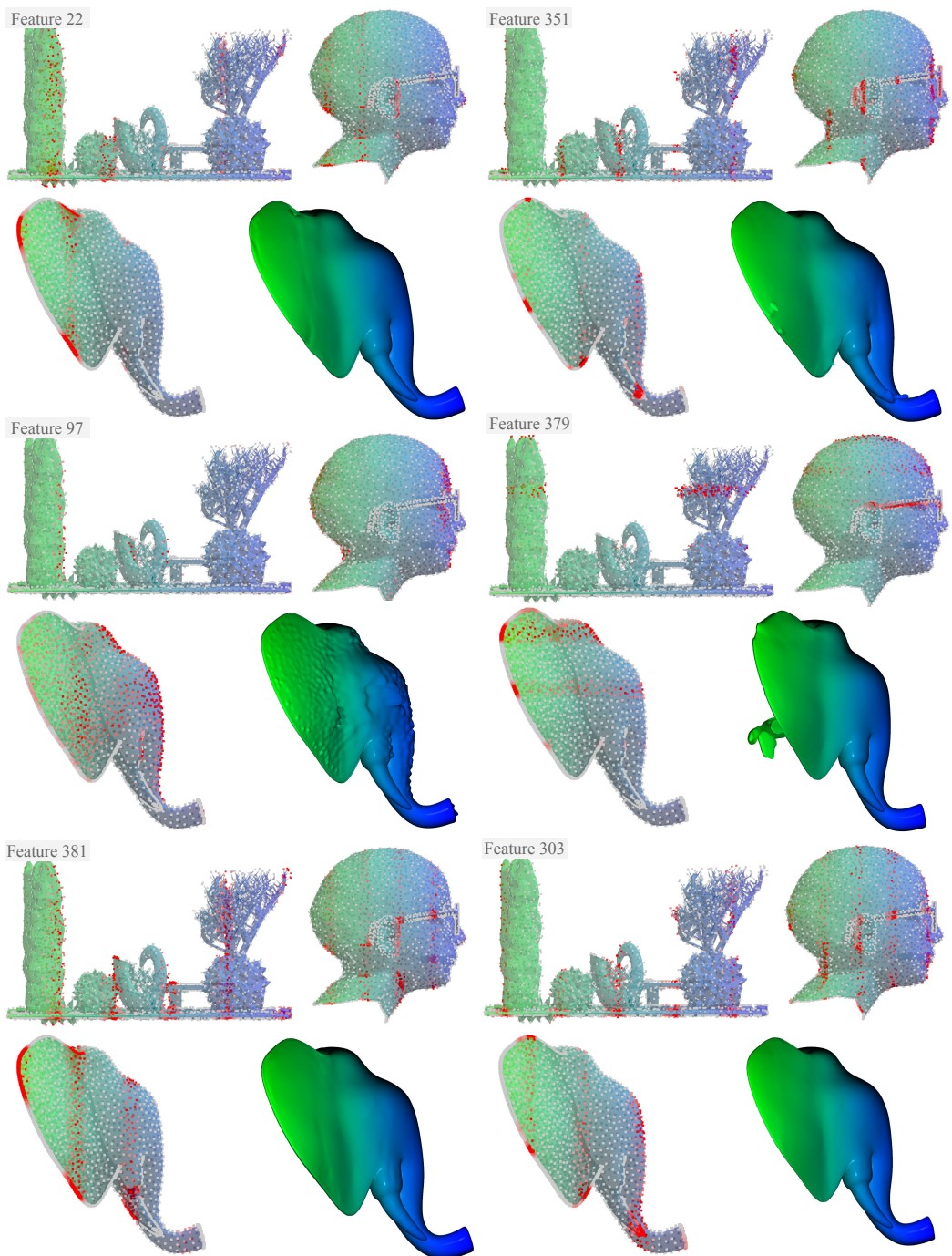

Figure 11: *Feature 22*: z-axis encoding. *Feature 351*: z-axis encoding. *Feature 97*: Right-facing smoothness. *Feature 379*: y-axis encoding. *Feature 381*: z-axis encoding. *Feature 303*: z-axis encoding.

