# OpenReview forum: "Features Emerge as Discrete States: The First Application of SAEs to 3D Representations"
_ICLR.cc/2026/Conference — ICLR 2026 Poster_

### Official Review · Reviewer_JL4Z · 2025-10-23

**Soundness:** 2
**Presentation:** 1
**Contribution:** 1
**Rating:** 2
**Confidence:** 3

**Summary:**

This is a poorly written paper that seems to claim to be the first study on the application of SAEs on 3D datasets and presents corresponding findings and interpretations of the learned features.

**Strengths:**

Unfortunately, given the current organization and writing of the main content in this paper, it is extremely difficult to identify any valuable insights for readers to learn.

**Weaknesses:**

(W1) The contributions of this paper are vague and poorly discussed. For example, in line 41, the author states that "the scope of data domains has been limited — recent feature interpretability studies have focused on discrete and structured data, like image and text, rather than continuous or unordered data, ..." However, I don't see why the solution to this would be to study SAE applications on 3D data, as the authors mentioned in lines 48–49 and highlighted in the abstract. Why don't the authors try continuous and unordered data in 1D and 2D spaces first? Besides, a contradictory and confusing point is that the 3D point cloud data used in this paper's experiments is also discrete.

(W2) The writing of the methods section is very confusing and difficult to follow. It seems that the author proposes a summary formula for SAE works in Eqn (2). However, first, it is not clear how the authors derive and denote the well-known representation learning formula of Eqn (1) into Eqn (2). Second, it is unclear how the authors convert the related SAE works into their formula. For example, in lines 113–114, the authors state that "Recent LLM studies (Cunningham et al., 2024) use a sparse autoencoder (SAE) to approximate this decomposition with the assumption that $\bf{α}$ is sparse." However, it seems that the SAE formula in Cunningham et al. (2024) has nothing to do with the left part of Eqn (3) of this paper, and their α is a hyperparameter in the final loss function to control the sparsity of the reconstruction. And it has a completely different meaning from the proposed set of scalars $\bf{α}$ in this paper. Similar confusing descriptions and links to related works are everywhere in the methods section. Lastly, it is unclear how the authors "apply" an SAE to Dora-VAE in their major 3D data application. What is FPS in Eqn (7)? Where are the proposed E and $\bf{α}$ notations in Section 3? How exactly do the authors modify a VAE framework using an SAE structure?

**Questions:**

In generanl, the authors are encouraged to re-setup the goal and scope of this research and rewrite the whole paper for the future submissions.

Please also respond and provide clear explanations for the confusions I described above, which may change my opinions.

---

> ### Author Response · Authors · 2025-11-14
> **W1 Response**
>
> Thank you for your time and efforts. Clearly, our presentation was imprecise, and we appreciate the opportunity to clarify. We refine our contributions and methods here, and are eager to discuss further areas for improvement. We have also revised the submission with your comments in mind. ***Edit: We label changes in our revised pdf. Changes aimed towards you are colored blue.***
>
> # Contributions
> Recent works in feature interpretability have focused almost solely on foundational models trained on textual data, which draws tokens from a finite, discrete vocabulary. We find this limits the field on two fronts. First, recent feature decomposition techniques, particularly SAEs, have rarely been applied to industries that use unordered data with continuous features. Second, we still lack a satisfactory explanation on how features are learned by models, a problem we think is exacerbated by the focus on ordered data drawn from a discrete set.
>
> **1) The first application of SAEs to 3D data. (Sections 2, 3)**
>
> We apply an SAE to the latent vectors of a 3D reconstruction model. This allows us to observe the features the model considers important when encoding 3D point cloud data. We then prove the veracity of these features by modifying the latent vectors and observing the decoded output. Identifying interpretable features is highly valuable for fields that frequently use AI tools on 3D data (animation, design, architecture, etc), as it greatly improves transparency.
>
> > Besides, a contradictory and confusing point is that the 3D point cloud data used in this paper's experiments is also discrete.
>
> The reviewer is right to point out our mistake in saying 3D point cloud data are not discrete — our language was imprecise. We mean that, while 3D point clouds are discretized for computation, the domain (position of geometry/points) is continuous. Conversely, text draws input tokens from a finite, discrete vocabulary.
>
> > I don't see why the solution to this would be to study SAE applications on 3D data.
>
> We choose to report on 3D point cloud data because learning structural features from an unordered, continuous domain is a qualitatively distinct challenge from the status quo. In addition, the behaviors observed later in our paper (discrete, state-like feature activations, Section 4) are striking because they arise from an entirely continuous domain. Finally, 3D point clouds are practical to study feature dynamics, because ***a)*** it is visually obvious when detected features have semantic meaning, ***b)*** existing datasets have a wide variety of immediately recognizable objects with unique semantic combinations ***c)*** many large industries rely on AI tools that use this domain, and ***d)*** we avoid datasets that are noisy, synthetic, homogenous, sparse, and/or constructed for toy experiments.
>
> > Why don't the authors try continuous and unordered data in 1D and 2D spaces first?
>
> We recognize the possibility of studying continuous and unordered 1D or 2D data. However, to the best of our knowledge, there is no well-known 1D or 2D domain that is continuous, unordered, semantically varied and cogent, and would practically benefit from interpretability studies to the same degree as 3D point cloud data. We do think the interpretability field can benefit from exploring such data domains, but it is a separate direction we leave for future work.
>
> **2) Detailed reports and analysis on unexpected behaviors in latent features. One primary example is, during feature ablation, the observation of discrete, state-like behavior in an otherwise continuous space. (Sections 4, 5)**
>
> As discussed in the introduction, and further in depth in Appendix A, we believe the field’s understanding of feature dynamics is lacking. Specifically, although other research has improved our techniques for extracting features, we have yet to establish a good theory for how or why these features were learned by the model. We report counter-intuitive feature behaviors that imply a complex set of learning dynamics.
>
> For example, Figure 2 and Section 4 describe the emergence of state-like positional features from data in a continuous space. This is highly anomalous — why might a model learn discrete positions when the input points are continuous? Any general theory on a model’s learning dynamics needs to address the observations we provide.
>
> **3) A novel framework that not only explains this behavior, but is potentially applicable to a wide range of models. (Section 2, explains 4, 5)**
>
> By rewriting the gradient update step (Equation 6) to include presence $\pmb{\alpha}$ and identity $\mathbf{E}$, we provide a framework that explains how our observed behaviors occur. We directly address the previously stated gap by proposing a theory on feature learning dynamics. This theory is potentially generally applicable, as it is derived solely from the gradient descent optimization step that is applicable to all neural network models.

---

> ### Author Response · Authors · 2025-11-14
> **W2 Response**
>
> # Methods
> When stated broadly, our methods are straightforward. We examine an existing 3D reconstruction model (Dora-VAE), which follows a simple encoder-decoder structure. We apply an SAE to the encoded latent vectors to observe what features Dora-VAE uses. Finally, we modify the latent vectors along learned directions in feature space and observe unexpected behaviors in the output.
>
> As we need to notate the feature decomposition formulas, the SAE architecture, the feature dynamics in the gradient step, and the Dora-VAE architecture, we appreciate that the notation is very dense. We are very willing to adjust the notation if it is then significantly easier to comprehend — please continue to indicate if a particular definition is confusing.
>
> ---
>
> > It seems that the author proposes a summary formula for SAE works in Eqn (2). However, first, it is not clear how the authors derive and denote the well-known representation learning formula of Eqn (1) into Eqn (2).
>
> **Equations 1 and 2** define the notation we use for feature representation in latent vectors, not the definition of an SAE. Equation 1 is a definition of a general neural network model as the composition of two functions, to isolate $f(\theta_f, \mathbf{x})$ as the latent vector. In Equation 2, we shorten the latent vector symbol to $\mathbf{z}$, and decompose $\mathbf{z}$ to the combination of feature vectors.
>
> This is a well-known representation learning formula — Bengio et al., (2013), Kim et al., (2018), and the cited papers on SAEs (see Appendix A) propose that latent vectors can be perceived as a linear combination of features. Borrowing the term from Kim et al., (2018), we define $E$ as the space spanned by these features, and $\mathbf{E}$ as the set of features. As a linear combination, each feature is modified by some scalar — we choose $\pmb{\alpha}$ to represent this vector of scalars and $\alpha_i$ as a single scalar, as it is a fairly common notation when describing linear combinations. We clarify these terms here.
>
> $$\mathbf{z} = f(\theta_f; \mathbf{x}) = \mathbf{E}(\theta_f)^T\pmb{\alpha}(\theta_f; \mathbf{x})$$
> $$\mathbf{E} = \\{ e_1, \ldots, e_n \\} \subset E$$
> $$\alpha= (\alpha_1, \ldots, \alpha_n)$$
>
> > And it has a completely different meaning from the proposed set of scalars 𝛂 in this paper.
>
> Note that our $\pmb{\alpha}$ has nothing to do with $\alpha$ in Cunningham et al. (2024) — we will replace the confusing reference with Bricken et al., (2023), which serves the same purpose in-context. We are happy to replace this term with another appropriate symbol, if the reviewer feels it is necessary.
>
> **Equation 3** defines the SAE architecture used in our experiments. This architecture is the standard, replicated across most current SAE literature (Bricken et al., 2023, Bussman et al., 2024). **Equation 4** adds the auxiliary loss defined in Appendix A.2 of Gao et al., (2025).
>
> **Equation 5** is the standard update step in gradient descent optimization. For **Equation 6**, we substitute the latent vector $\mathbf{z}$ with our definition from Equation 2, and expand the gradient with product rule.
>
> **Equations 7 and 8** restate the encoding pipeline of Dora-VAE from Chen et al. (2024). We understand this description is extremely dense, but we want to avoid simply repeating work discussed in their publication, since we only discuss the encoded latent vectors. We primarily focus on $\mathbf{C}$, the set of $M$ pre-KL network states encoded from an input point cloud. Through KL, these states are each embedded to a latent vector $\mathbf{z}$. The SAE is applied to these latent vectors.
>
> If one requires the full architecture Dora-VAE operates on, we also invite them to examine Zhang et al. (2023) (https://arxiv.org/pdf/2301.11445), which Dora-VAE was predicated on.

---

> > ### Author Response · Authors · 2025-11-14
> > **Further Questions and Citations**
> >
> > > Similar confusing descriptions and links to related works are everywhere in the methods section.
> >
> > We try to address the reviewer’s stated confusions here, but please note that our chosen notation cannot be completely identical to all referenced literature. Several publications use the same symbol for two separate meanings; for example, Gao et al., (2025) uses $e$ for the reconstruction from dead latents, while Kim et al. (2018) uses it for feature identity. In addition, some symbols must be notated differently to address the change in scope; for example, Bussmann et al., (2024) uses $\mathbf{x}$ to represent the latent activations fed to the SAE, because the paper does not define the model on which they apply the SAE. On the other hand, we do define this model to describe the learning dynamics, so we use $\mathbf{x}$ to represent the inputs to the model, and $\mathbf{z}$ to represent the latent activations.
> >
> > > What is FPS in Eqn (7)?
> >
> > $\text{FPS}$ is the furthest point sampling algorithm Dora-VAE uses to select a subset from the input point cloud. This is built inside the Dora-VAE encoder; we only notate it to faithfully define the architecture.
> >
> > > Where are the proposed E and 𝛂 notations in Section 3?
> >
> > $\mathbf{E}$ and $\pmb{\alpha}$ are not included in Section 3 because they only pertain to feature dynamics — the gradient step of the model is rewritten in these terms in Section 2. Section 3 only defines Dora-VAE.
> >
> > > How exactly do the authors modify a VAE framework using an SAE structure?
> >
> > We do not modify the VAE architecture or weights for this paper. We only train the SAE on Dora-VAE’s encoded latent vectors, then modify these latents to observe the change in the decoder output.
> >
> > ---
> >
> > We hope this first rebuttal addresses your initial questions. Please let us know of any further areas needing clarification during this period — we are fully committed to improving the paper in your eyes. We hope to receive your feedback soon, as we are eager to improve off of your critiques.
> >
> > ---
> >
> > Bengio et al., 2013: “Representation learning: A review and new perspectives.”
> >
> > Bricken et al., 2023: “Towards monosemanticity: Decomposing language models with dictionary learning.”
> >
> > Bussmann et al., 2024: “Batchtopk sparse autoencoders.”
> >
> > Chen et al., 2024: “Dora: Sampling and benchmarking for 3d shape variational autoencoders.”
> >
> > Cunningham et al., 2024: “Sparse autoencoders find highly interpretable features in language models.”
> >
> > Gao et al., 2025: “Scaling and evaluating sparse autoencoders.”
> >
> > Kim et al., 2018: “Interpretability beyond feature attribution: Quantitative testing with concept activation vectors (tcav).”

---

### Official Review · Reviewer_nEVZ · 2025-10-28

**Soundness:** 3
**Presentation:** 2
**Contribution:** 4
**Rating:** 6
**Confidence:** 3

**Summary:**

This work provides the first application of Sparse Autoencoders (SAEs) to 3D data, decomposing the internal representations of a 3D Variational Autoencoder (VAE). The paper's central finding is that the model learns features as discrete states rather than continuous values, which are activated via phase transitions. This claim is supported by a large-scale analysis of 848k feature ablations, which reveal that: (1) High-impact features have a unimodal (single-peak) distribution of transition points. (2) Low-impact features have a bimodal (two-peak) distribution of transition points. The authors hypothesize this bimodality is evidence that the model actively manipulates superposition interference at inference time, pushing interference onto low-impact features to preserve the saliency of high-impact ones.

**Strengths:**

- Interpretability for 3D data is underexplored and interesting!
- Lots of experiment runs (e.g., 848k feature interventions) which makes the results robust.
- The theoretical explanation of how models see features as a decomposition of presence and identity is also interesting.
- The bimodal experiments (Fig 5) and visualization (Fig 3) are insightful.
- Validating the threshold t with max slope experiments is really nice!

**Weaknesses:**

- The learning dynamics and 3D contributions seem completely disjoint (although both interesting). Moreover, the paper makes broad claims about "a generally applicable, state-based feature framework." However, all the evidence is derived from a single model architecture (Dora-VAE) on a single data modality (3D point clouds). It's impossible to know if these findings (especially the bimodal transitions) are a fundamental property of feature learning, or a specific quirk of the Dora-VAE architecture e.g., and its cross-attention mechanisms.
- I am having difficulty understanding Figure 6 and more generally, the explanation for the bimodality. I generally get the intuition that the model would put more interference with the low-impact concepts, but have difficulty following the explanation in Lines 459-475.

Small stuff

- Figure captions should be more detailed and explain (i) why this plot is being shown and (ii) what takeaways the reader should make. Not just a short description of what is shown.

**Questions:**

- Do these findings hold across different models and data types? Or is this only for Dora-VAE and 3D pointcloud data? Showing these findings hold on another model/domain would increase the contribution strength for the learning dynamic theory section a lot. Or, this could be added as a weakness/limitation to the paper since it is only applied to 3D data (even though applying SAEs to 3D data is a contribution itself, it does not mean all general findings on 3D data will hold to other domains).
- What is Figure 6 showing? It is not clear if it is a real experiment or just a visual aid. It is just mentioned in the first line of Line 454 and 459, but the text and caption are insufficient to parse the figure.
- Moreover, could the authors kindly explain in more clear terms how the bimodal transition arises? I think this section could even be dropped, or written as a 'hypothesis' rather than making stronger claims about superposition/interference.

---

> ### Author Response · Authors · 2025-11-19
> **Pointwise Response**
>
> Thank you very much for the positive feedback. We have revised our paper based on your suggestions, and have published the first revision. ***We have labelled changes targeted towards you in teal.*** We also summarize our improvements below, and mark the lines in which corrections appear.
>
> ### Weakness 1:
>
> > The learning dynamics and 3D contributions seem completely disjoint.
>
> We would argue that the joint presentation of 3D contributions and learning dynamics is necessary. Our research targets two gaps — the lack of interpretability investigations into 3D data, and the lack of discussion on feature learning dynamics. However, we make the claim that the latter is caused by the former; by restricting the domains of feature decomposition research to mostly text, and generally, ordered data, we fail to study feature behaviors that inform a more general theory of learning dynamics. *Lines 43-49 and 82-84* clarify this direction (although were targeted at reviewer JL4Z).
>
> >  Moreover, the paper makes broad claims about "a generally applicable, state-based feature framework."
>
> This is a fair criticism, and we have significantly narrowed our framing as a result. We do not intend to claim a fully proven, universal framework. Rather, we provide a novel exploration of feature learning dynamics, with an initial mechanism to guide further research.
>
> The scope of a general, ground-up framework is substantial, given our current limited understanding of feature learning. We instead aim to report on anomalous behaviors and mechanisms that build our intuition of the field; we see these findings as important groundwork for developing a universal theory.
>
> >  All the evidence is derived from a single model architecture on a single data modality.
>
> Since our submission to ICLR, Gurnee et al. (2025) has published research observing discrete phase transitions in text, similar to our work. We are delighted to see some of our observations replicated in other domains.
>
> We agree that other modalities and models are exciting areas of study. However, performing a cross-model investigation with this level of detail is substantial in scope, so we focus on establishing and describing our observed behaviors with one model. In regards to modalities, our goal was to study domains that have been overlooked in favor of text, so we restrict ourselves to 3D data. Keeping our scope specific allows us to analyze and report on observed behaviors with the necessary diligence.
>
> We do intend to explore other methods and domains, in order to further confirm the results we observe, and outline a future works section. *Lines 523-531*
>
> ### Weakness 2:
>
> > I am having difficulty understanding Figure 6 and more generally, the explanation for the bimodality.
>
> To address this, we have completely rewritten Section 5.2, Figure 6, and Figure 7. Figures 6 and 7 are purely visual aids to illustrate the bimodality mechanism and redistribution of interference, respectively. In addition, we have modified our language to make the speculative nature of the claim more clear. *Lines 476-513*
>
> ### Small Stuff: Figure Captions
>
> We have updated all figure captions to be more independently understood.
>
> ### Questions
>
> We believe we have addressed all questions in the ‘weaknesses’ section. However, we are eager to improve the paper — if you have any further suggestions or queries, please let us know.
>
> ---
>
> Gurnee et al., 2025: “When Models Manipulate Manifolds: The Geometry of a Counting Task.” https://transformer-circuits.pub/2025/linebreaks/index.html#char-count

---

> ### Comment · Reviewer_nEVZ · 2025-11-26
>
> I am still not convinced that 3D and learning dynamics need to be presented together. There are a variety of works that train SAEs on images (for example [a,b,c]) and I don't see why the learning dynamics for these SAEs wouldn't be interesting to explore before jumping to 3D? Additionally, all of these citations should be included in the related work. Especially regarding the universality discussion and [a].
>
> I appreciate the reworking of Section 5.2 and the figures. I know you are running out of space, but the Figure 7 caption should be updated to be self-contained and more descriptive. The idea of the model moving interference to less-important features is an interesting hypothesis nonetheless and further showing evidence of this would be quite interesting.
>
> I also looked at the other reviews. I see that they also highlight the issues with framing of 3D + dynamics being necessarily tied and the study being solely on a single model + dataset. It is still not clear to me whether the core findings in the paper are solely for this model, 3D data in general, or could apply to multiple models + modalities. I agree though, that the findings are interesting enough to potentially warrant acceptance. For these positive and negative reasons, I will keep my score as is.
>
> [a] Universal Sparse Autoencoders: Interpretable Cross-Model Concept Alignment (ICML 2025)
>
> [b] Discover-then-Name: Task-Agnostic Concept Bottlenecks via Automated Concept Discovery (ECCV 2024)
>
> [c] Archetypal SAE: Adaptive and Stable Dictionary Learning for Concept Extraction in Large Vision Models (ICML 2025)

---

> ### Author Response · Authors · 2025-11-26
>
> We thank you, again, for fruitful feedback. Regardless of the final decision, we authors have a better understanding of how to frame future publications. We will include the cited publications and the caption of Figure 7 before the end of the rebuttal period.
>
>  > The learning dynamics and 3D contributions seem completely disjoint.
>
> We argue that our paper should not be evaluated based off of the potential of other possible directions. Instead, we want to emphasize how the two sections are mutually beneficial. If we focused on learning dynamics through work in image feature decomposition, a significant portion of our work would replicate known behavior. If we removed the learning dynamics and focused on only 3D feature decomposition, we reduce our work to cataloging features.
>
> If we might summarize your argument (and please correct us if we are wrong), you are suggesting that the contribution/quality of the paper would be higher if we either **a)** studied learning dynamics by building off of existing work about SAEs applied to image data, or **b)** focused on the feature investigation of 3D data, without addressing learning dynamics. We respectfully disagree.
>
> **a)** If we studied learning dynamics based off of the existing body of research in image data, we believe our contribution would have less value. This is not to say that studies of learning dynamics in image models is uninteresting. On the contrary, we are excited to address this area ourselves in the future, and acknowledge the benefits provided by existing work. But if we did so here, our paper would contribute less - a significant portion of our work would confirm existing observations of feature decomposition in image data. In contrast, we provide the first application of SAEs to 3D data, and thus a new understanding of how a reconstruction model organizes features like position and texture. Previously, researchers in the field understood generally how an image model segments concepts (as by the studies [a,b,c]); they did not know how a 3D model did it.
>
> **b)** If we solely focused on this feature investigation of 3D data, however, we would contribute less to the intuition behind feature dynamics. We find that the feature interpretability field has focused too much on what features are discovered, rather than why these features were learned. Although cataloging these features is important, we see a growing need to move towards explanatory insight. So, after we discovered how a 3D reconstruction model organizes features of position and texture, we choose to investigate why the model learned these features in such a manner. By providing an explanation as to why certain features were learned, rather than only observation, we distinguish our work.
>
> We acknowledge that examining learning dynamics through image data could yield important intuition on a universal framework. However, our paper should not be judged based off of the merits of other potential work. We believe our paper still has significant value: the examination of domain that is under-explored, with an approach that is underrepresented. We hope our rebuttal clarifies our motivations and contributions.

---

### Official Review · Reviewer_y1WU · 2025-11-03

**Soundness:** 3
**Presentation:** 1
**Contribution:** 3
**Rating:** 4
**Confidence:** 1

**Summary:**

This paper investigates the internal features of a 3D reconstruction model by being the first to apply Sparse Autoencoders (SAEs) to the 3D domain. The authors use an SAE to analyze the latent vectors of a Dora-VAE (a 3D reconstruction VAE) trained on the Objaverse dataset.  The central finding is that the Dora-VAE's internal features are **discrete, not continuous**.

This paper proposes a novel theory framework oo explain "unintuitive" behaviours of VAEs. They analyze the gradient and decompose it into two components: the feature's **"presence** ($\alpha_j$) and its **"identity"** ($e_j$). The framework suggests that the model prefers to learn discrete features because the learning signal for a feature's *identity* is scaled by its *presence*. Specifically, the **unimodal** (single-peaked) distribution of transition points for **high-impact features** is explained by the feature's "presence" term. In contrast, the **bimodal** (two-peaked) distribution observed for **low-impact features** is explained by the model's active manipulation of superposition. The authors further apply SAEs into VAEs for 3D data and explain the unusual properties.

**Strengths:**

1. New Theoretical Framework: The decomposition of the learning gradient into "presence" ($\alpha_j$) and "identity" ($e_j$) provides a new and powerful lens for why features emerge in a certain way, moving beyond just observing what features are learned.
2. Well-motivated. The application of SAEs to the 3D domain is motivated by the theory analysis to extend interpretability.
3. Strong Explanatory Power: The proposed framework successfully connects several counter-intuitive empirical observations (discrete features, sigmoidal loss curves, and bimodal transition points) into a single, cohesive theory.

**Weaknesses:**

1. The writing can be improved for clarity, such as the definition of ARC and loss difference. It is not clear how to calculate them. What the colored points representing are also not depicted.
2. The findings are based on the Dora-VAE architecture, which processes 3D models by sampling points and using point features and positional encodings. The introduction of SAE is an incremental contribution.
3. The paper uses the terms "discrete state space" and "phase transition" heavily. While this is a helpful analogy for the observed sigmoidal loss curves, it may overstate the case.

**Questions:**

How important it is for the application of SAE to 3D data? What is performance of the proposed method in Dora-bench?

---

> ### Author Response · Authors · 2025-11-19
> **Pointwise Response**
>
> Thank you very much for your feedback. Based on your review, we have improved the presentation of our paper — if you examine our current revision, ***we have labelled changes targeted towards you as red***. We also summarize our improvements below, and mark the lines in which corrections appear.
>
> ### Weakness 1:
>
> > The writing can be improved for clarity, such as the definition of ARC and loss difference. It is not clear how to calculate them.
>
> We have tightened and clarified our definitions, particularly in sections 4 and 5. We replace the vague phrasing ‘loss difference’ with **impact**, or $\Delta L$. Recall that every decoded reconstruction during ablation is compared to the original object with MSE loss. We define the impact of an ablation as the difference in MSE between when no ablation has occurred ($t=0$) and when the ablation is completed ($t=1$). We use impact to measure how much an ablation affected the reconstruction, i.e. how important the ablated feature was. *Lines 354-358*
>
> The ablation-response curve **(ARC)** is a type of plot we use to visualize and compare ablations by plotting normalized MSE against $t$-values. We normalize such that MSE is 0 when $t=0$ and 1 when $t=1$. This allows us to observe general trends amongst ARCs, such as discretization and transition point. *Lines 361-365*
>
> > What the colored points representing are also not depicted.
>
> We assume you are referring to the points in Figure 2, which we have also clarified. For each encoded 3D object, we obtain a set of latents $\\{\mathbf{z}\_i\\}\_{i=1}^M$, and $\pmb{\alpha}\_i$ for each latent by passing it through the SAE. We plot the $M$ latents as points from their initially sampled positions $\mathbf{P_d}$. Finally, to examine a feature $j$, we color each point $i$ of each latent based off the presence $\alpha_{i,j}$. *Lines 242-246*
>
> This allows us to visualize how features are organized in each 3D object. We highlight some features that attend to the position of a point in space — notably, they are not a continuous range of values, but instead divide the object into discrete regions. *Lines 236-240*
>
> ### Weakness 2:
>
> > The findings are based on the Dora-VAE architecture… The introduction of SAE is an incremental contribution.
>
> We apologize for the confusion, but the findings of this paper were not based on the Dora-VAE architecture. Our contributions center on decomposing the latent space of a 3D reconstruction model to study feature dynamics, not to improve the performance of the model’s reconstruction.
>
> Although the model’s performance is impressive in its own right, its latent state is a black box — the information carried by latent vectors is not visible to humans, so we cannot determine how Dora-VAE encodes features efficiently. Only with the SAE, an interpretability technique that examines these latent vectors, can we visualize how concepts such as position, texture, and non-occupancy are stored. We modify *Figure 1* to address this confusion.
>
> Beyond interpreting Dora-VAE’s latent vectors as features, we also perform numerous feature ablations and additions to examine the general effect of features on the reconstruction. We record unexpected effects for posterity, and propose a framework that explains these behaviors. The application of the SAE for feature decomposition is the basis behind these contributions.
>
> ### Weakness 3:
>
> >  The paper uses the terms "discrete state space" and "phase transition" heavily. While this is a helpful analogy for the observed sigmoidal loss curves, it may overstate the case.
>
> Thank you for pointing this out. We authors are also cautious of overusing this language, and we have tempered our terminology in the latest revision. Specifically, we want to avoid claiming the feature space is discrete — instead, we argue that the tendency of features to avoid middling presences encourages a state-like distribution.
>
> However, we also agree that describing the behavior as approximating a state space is a helpful intuition for understanding the feature dynamics. We aim to balance clarity for the reader and overclaiming.
>
> ### Question 1:
>
> Addressed in Weakness 2.
>
> ### Question 2:
>
> > What is performance of the proposed method in Dora-bench?
>
> Unfortunately, the time and resources to qualitatively examine Dora-bench for learning dynamics are outside the scope of this rebuttal. We feel Objaverse adequately represents Dora-bench, however, as it makes up a significant portion of the benchmark.
>
> Objaverse not only makes up 35% of Dora-bench, but also represents >90% of the objects in Dora-bench’s highest classification for geometric complexity (Chen et al., (2024)). Thus, we feel that our observations of Dora-VAE on Objaverse will be reflected in Dora-bench as well.
>
> ---
>
> We sincerely thank you for your time, and are looking forward to further suggestions to improve the paper.
>
> ---
>
> Chen et al., 2024: “Dora: Sampling and benchmarking for 3d shape variational autoencoders.”

---

> > ### Comment · Reviewer_y1WU · 2025-11-24
> >
> > The main contribution is to interpret the role of each feature. However, the existing evidence is self-interpret by $\alpha$. More visualisation or extra evaluations are needed to assess the correctness of  the proposed interpretation in Figure 2.
> >
> > For example, how about feature permutation? What it would the reconstruction look like for the examples in Figure 2 by by permutating one feature, e.g., 166. Will the error parts match the red points?
> >
> > And more examples should be presented in Appendix.

---

> ### Author Response · Authors · 2025-11-26
> **Additional Examples**
>
> We are happy to provide further examples that each individual feature has semantic meaning, and do so in our revised *Appendix C*. We select another 18 features that are shared between the three objects in Figure 2 and similarly highlight them. We also permute each feature for one object to demonstrate causality (as in *Section 3.2* and *Figure 3*). Note that ablations do not always warp the object, due to the redundancy among latents discussed in Appendix B.2, section ⓓ.
>
> We also agree that feature permutation is another way to confirm the correctness. We draw the reviewer’s attention to *Section 3.2* and *Figure 3*, which demonstrate feature permutations (we notate as ablations) that modify the reconstruction. The changes in reconstruction match perfectly with the red points, and do so across all examined features and 3D objects. In addition, if the reviewer is interested in a quantitative analysis of feature permutation, with measures of the MSE of the reconstruction, we do this in *Section 4*.
>
> We also want to clarify — although assessing the role of each feature is an important contribution, we feel the main contribution is the discussion on feature learning dynamics, and how the unique properties of 3D inform our intuition of feature decomposition. Thus, we are curious as to the reviewer’s thoughts on our earlier rebuttal. Has the response and respective changes addressed your previous concerns?

---

### Author Response · Authors · 2025-11-19
**Rebuttal Revision**

Thank you, reviewers, for your comments. We have responded to each of your suggestions, and have posted a revised version of paper significantly improved by your comments.

**We have colored specific changes for each reviewer:**

**Reviewer 1: Red**

**Reviewer 2: Teal**

**Reviewer 3: Blue**

We are looking forward to further comments.

---

### Author Response · Authors · 2025-12-01
**Public Summary of Rebuttal**

We very much appreciate the discussion on our paper. We understand that the chosen reviewers had difficulty with our presentation, and, as such, we have drastically improved the clarity of our contributions and methods, and performed further experiments to demonstrate the veracity of our observations. It is disappointing that the circumstances do not allow our reviewers to engage in further discussion, as we believe we have fully addressed their stated concerns.

However, we believe the merits recognized by the reviewers still stand strong. Our paper addresses two underrepresented niches in interpretability research: **interpretability for 3D data** and **feature learning dynamics discussion**. We are happy to see that reviewers appreciate the strong explanatory power provided, perceiving our methodology as robust and our presented framework as insightful.

We thank our reviewers for their time, knowing that with their suggestions, future readers will better understand our paper's contributions. We hope the AC appreciates our comprehensive responses to reviewer questions.

---

### Meta-Review · Area_Chair_2nut · 2026-01-06

**Summary:**

Given that the reviewers' confidence scores are relatively low, I have conducted a careful reading of the paper. The work presents an interesting approach to analyzing latent space, addressing a notable direction in the field of 3D representations and generative models. I believe this paper would be a valuable contribution to the community.

However, a primary limitation is that the analysis is restricted to Dora-VAE (which is VecSet-based). Other 3D generative models that utilize latent spaces, such as LION, XCUBE, and TRELLIS, could also be relevant. Discussing these models would significantly enhance the proposed theory. Overall, I recommend acceptance, provided that the authors incorporate the promised changes from the rebuttal.

**Reviewer Concerns:**

The Authors are generally more persuasive in some points, e.g.,
1. The contribution (as a first steps in the domain of interpretable representations)
2. The clarity of the writing (more contents are in the revised paper)

**Reviewer Scores:**

Only the reviewer was not be able to participate the discussion. Overall, I believe the authors did a great job in clarifying the paper.

---

### Decision · Program_Chairs · 2026-01-26

Accept (Poster)